# Acute Pulmonary Embolism in Patients with and without COVID-19

**DOI:** 10.3390/jcm10102045

**Published:** 2021-05-11

**Authors:** Antonin Trimaille, Anaïs Curtiaud, Kensuke Matsushita, Benjamin Marchandot, Jean-Jacques Von Hunolstein, Chisato Sato, Ian Leonard-Lorant, Laurent Sattler, Lelia Grunebaum, Mickaël Ohana, Patrick Ohlmann, Laurence Jesel, Olivier Morel

**Affiliations:** 1Division of Cardiovascular Medicine, Nouvel Hôpital Civil, Strasbourg University Hospital, 67000 Strasbourg, France; antonin.trimaille@chru-strasbourg.fr (A.T.); anais.curtiaud@chru-strasbourg.fr (A.C.); matsuken_22@yahoo.co.jp (K.M.); benjaminmarchandot@gmail.com (B.M.); jean-jacques.vonhunolstein@chru-strasbourg.fr (J.-J.V.H.); okoge16@gmail.com (C.S.); patrick.ohlmann@chru-strasbourg.fr (P.O.); laurence.jesel@chru-strasbourg.fr (L.J.); 2INSERM (French National Institute of Health and Medical Research), UMR 1260, Regenerative Nanomedicine, FMTS, 67000 Strasbourg, France; 3Radiology Department, Nouvel Hôpital Civil, Strasbourg University Hospital, 67000 Strasbourg, France; leonard.lorant@gmail.com (I.L.-L.); mickael.ohana@chru-strasbourg.fr (M.O.); 4Haematology and Haemostasis Laboratory, Centre for Thrombosis and Haemostasis, Nouvel Hôpital Civil, Strasbourg University Hospital, 67000 Strasbourg, France; laurent.sattler@chru-strasbourg.fr (L.S.); lelia.grunebaum@chru-strasbourg.fr (L.G.)

**Keywords:** COVID-19, SARS-CoV-2, thrombosis, acute pulmonary embolism, inflammation, computed tomography pulmonary angiography

## Abstract

Introduction. Acute pulmonary embolism (APE) is a frequent condition in patients with COVID-19 and is associated with worse outcomes. Previous studies suggested an immunothrombosis instead of a thrombus embolism, but the precise mechanisms remain unknown. Objective. To assess the determinants and prognosis of APE during COVID-19. Methods. We retrospectively included all consecutive patients with APE confirmed by computed tomography pulmonary angiography hospitalized at Strasbourg University Hospital from 1 March to 31 May 2019 and 1 March to 31 May 2020. A comprehensive set of clinical, biological, and imaging data during hospitalization was collected. The primary outcome was transfer to the intensive care unit (ICU). Results. APE was diagnosed in 140 patients: 59 (42.1%) with COVID-19, and 81 (57.9%) without COVID-19. A 812% reduction of non-COVID-19 related APE was registered during the 2020 period. COVID-19 patients showed a higher simplified pulmonary embolism severity index (sPESI) score (1.15 ± 0.76 vs. 0.83 ± 0.83, *p* = 0.019) and were more frequently transferred to the ICU (45.8% vs. 6.2%, *p* < 0.001). No difference regarding the most proximal thrombus localization, Qanadli score (8.1 ± 6.9 vs. 9.0 ± 7.4, *p* = 0.45), the proportion of subsegmental (10.2% vs. 11.1%, *p* = 0.86), and segmental pulmonary embolism (35.6% vs. 24.7%, *p* = 0.16) was evidenced between COVID-19 and non-COVID-19 APE. In COVID-19 patients with subsegmental or segmental APE, thrombus was, in all cases (27/27 patients), localized in areas with COVID-19-related lung injuries. Marked inflammatory and prothrombotic biological markers were associated with COVID-19 APE. Conclusions. APE patients with COVID-19 have a particular clinico–radiological and biological profile and a dismal prognosis. Our results emphasize the preeminent role of inflammation and a prothrombotic state in these patients.

## 1. Introduction

Since the outbreak of coronavirus disease 2019 (COVID-19), acute pulmonary embolism (APE) has been recognized as a frequent complication that carried a dismal prognosis [1,2]. Evidence from autopsy series reported endotheliitis [3], pulmonary vascular microthrombosis [4], and intense inflammation within the pulmonary vasculature. Laboratory features of hypercoagulability in COVID-19 include marked elevation of D-dimer and fibrinogen level, and the presence of lupus anticoagulant (LA) at high frequency [5,6,7]. The concept of immunothrombosis has emerged to describe radiological findings of subsegmental or segmental thrombi [8]. Moreover, a decline in incidence of APE and imaging procedures has been reported due to the COVID-19 pandemic [9].

In an effort to better characterize the pathological mechanisms underlying COVID-19 associated APE, we sought to compare COVID-19 and non-COVID-19 APE. The objectives of our study were to compare the difference regarding clinical, biological, and radiological characteristics of COVID-19 and non-COVID-19 patients with APE admitted to the general ward at our institution, and to assess the impact of COVID-19 in APE prognosis.

## 2. Methods

### 2.1. Setting and Study Population

All patients with clinical and/or biological suspicion of APE (D-Dimers elevation) and computed tomography (CT) pulmonary angiography (CTPA) performed in the Radiology departments of Strasbourg University Hospital (two centers: Nouvel Hôpital Civil and Hôpital de Hautepierre) were reviewed for imaging techniques and findings. We retrospectively included all patients ≥18 years of age with CTPA confirmed APE admitted to general wards between 1 March and 31 May of the years 2019 and 2020. Exclusion criteria were: an unclear diagnosis of APE or the absence of APE on CTPA. Anonymized data from all patients were collected via our institution’s electronic health record. A complete set of clinical, biological, and imaging data were recorded. The present study was approved by the research ethics committee of Strasbourg Hospital (authorization CE-2020-57) who waived the need of informed consent.

Medical management was left to the discretion of the treating physician. All APE patients were treated with therapeutic anticoagulation according to current guidelines [10]. Non-systematic thromboprophylaxis before the index APE event included standard doses (subcutaneous Enoxaparin at 0.4 mL per day, subcutaneous Fondaparinux at 2.5 mg per day, or intravenous unfractionated heparin at 200 IU per hour) or an intermediate dose (subcutaneous Enoxaparin at 0.4 mL twice per day).

### 2.2. Study Definitions

A confirmed case of COVID-19 was defined by a positive result of a reverse-transcriptase–polymerase-chain-reaction (RT-PCR) assay of a specimen collected on a nasopharyngeal swab and by typical findings of COVID-19 at chest CT (bilateral and peripheral ground glass opacities and/or alveolar consolidations) [11]. Venous thromboembolism (VTE) risk was assessed on admission to hospital via the Padua Prediction Score and the IMPROVE score [12,13,14,15]. The simplified pulmonary embolism severity index (sPESI) score, a risk stratification tool to determine the mortality of patients with newly diagnosed APE, was calculated for every patient at the time of APE diagnosis [16].

### 2.3. Imaging

APE was diagnosed either at the time of admission or during hospital stay by a CTPA acquired on a 64-row or greater scanner, after injection of 50 to 75 mL of high concentration iodine contrast media. The most proximal localization of thrombus in the pulmonary vasculature was recorded. Total thrombus load was assessed via the Qanadli CT pulmonary obstruction index and calculated on CTPA for all patients [17]. Deep vein thrombosis (DVT), in subjects with signs or symptoms, was diagnosed by a complete duplex ultrasound from thigh to ankle with Doppler waveforms and images.

Lesions severity of COVID-19 at CT was visually classified following the European Society of Radiology (ESR)/European Society of Thoracic Imaging (ESTI) guidelines [11]. The extent of COVID-19 disease on chest CT scans was assessed as pulmonary injuries extension in percentage of the total pulmonary field and classified as minimal (stage 1 < 10%), moderate (stage 2 = 10 to 25%), severe (stage 3 = 25 to 50%), and critical (stage 4 > 50%).

### 2.4. Laboratory Tests

Laboratory values included leukocytes count, platelets count, hemoglobin, and inflammatory markers, such as C-reactive protein (CRP). Hemostasis assays (fibrinogen, D-dimer, lupus anticoagulant (LA) detection) were analyzed on STA-R^®^ Evolution (Diagnostica Stago^®^, Asnières-sur-Seine, France) with standard commercial reagents and protocols.

LA detection was based on several tests. First, two screening tests were performed, respectively a Diluted Russel Viper Venom Time (dRVVT screen) made with the STA^®^-Staclot dRVV Screen reagent (Stago), and an activated partial prothromboplastin time (aPTT) performed with the STA^®^-PPT A reagent (Stago). Positivity of one or both screening tests induced a mixing test at 1:1 proportion with a commercial frozen PNP (CRYOcheck™ Pooled Normal Plasma, CRYOcheck, Montpellier, France). Moreover, a positive dRVVT screen induced a confirmatory test with an increased concentration of phospholipids (dRVVT confirm), performed with the STA^®^-Staclot dRVV Confirm reagent (Stago). dRVVT screen, DRVVT confirm, and aPTT results were expressed as a ratio of patient-to-PNP. Mixing tests results were expressed as an index of circulating anticoagulant (ICA). LA was considered as positive only if the normalized dRVVT ratio (screen ratio/confirm ratio) was >1.2 and all causes of false positive were excluded (i.e., anticoagulation conditions).

### 2.5. Study Outcomes

The primary outcome was transfer to the intensive care unit (ICU), which reflected the need for hemodynamic and respiratory support. Secondary outcomes included in-hospital death and the need of mechanical ventilation during hospital stay.

### 2.6. Statistical Analysis

Continuous variables were expressed as mean ± standard deviation or median and interquartile range as appropriate and categorical variables as counts and percentages. Continuous variables were compared with the use of parametric (Student’s *t* test) or non-parametric Mann–Whitney tests as appropriate. Categorical variables were compared with chi-square test or Fischer’s exact test. The time to event was defined as the time from hospital admission to the date of transfer to the ICU, with patients censored at death or end of study. The impact of COVID-19 on prognosis during APE was assessed using both univariate and multivariate Cox hazard model.

A two-tailed *p* value < 0.05 was considered significant. Statistical analyses were performed using SPSS 17.0 for Windows (SPSS Inc., Chicago, IL, USA).

## 3. Results

### 3.1. Baseline Characteristics

A total of 8722 chest CT were performed during the 2020 period (from 1–31 March), with a 915% and 12% increase, respectively, in chest CT (*n* = 3573) and CTPA (*n* = 808) compared with the equivalent period in 2019 (Figure 1).

APE was diagnosed in 140 patients: 59 patients with COVID-19 and 81 without COVID-19. A total of 8 non COVID-19 related APE was registered during the 2020 period corresponding to an 812% reduction as compared with 2019. APE was diagnosed in median 3 (interquartile range (IQR) 10) days after admission for COVID-19 patients, and 0 (IQR 1) days for non COVID-19 patients. Imaging test for DVT was performed in 14/59 (23.7%) patients with COVID-19 and in 63/81 (77.8%) in patients without COVID-19. COVID-19 patients were younger, more frequently male and obese (Table 1). Traditional VTE risk factors such as previous VTE, previous APE and active cancer were less frequently encountered in patients with COVID-19 (*p* < 0.05). Specific VTE risk assessment models via the Padua prediction score and the IMPROVE score, were not significantly different between the two groups. The mean sPESI score was higher in patients with COVID-19 (1.15 ± 0.76 vs. 0.83 ± 0.83, *p* = 0.019) and a lower proportion of these patients had a low sPESI risk (16.9% vs. 33.3%, *p* = 0.034) (Table 2). Significantly more patients with COVID-19 were under standard or intermediate dose of thromboprophylaxis before APE in comparison with patients without COVID-19 (Table 1).

### 3.2. Imaging of Acute Pulmonary Embolism

No difference regarding the most proximal thrombus localization within the pulmonary vasculature could be evidenced between COVID-19 and non COVID-19 APE (Table 2). The proportion of subsegmental (10.2% vs. 11.1%, *p* = 0.86) and segmental pulmonary embolism (35.6% vs. 24.7%, *p* = 0.16) was not significantly different in patients with and without COVID-19. Among COVID-19 patients with subsegmental or segmental APE, thrombus was in all cases (27/27 patients) localized in areas with COVID-19 related lung injuries (Figure 2). Total thrombus load assessed via the Qanadli score did not significantly differ between the two subsets of patients (8.1 ± 6.9 in COVID-19 patients vs. 9.0 ± 7.4 in non COVID-19 patients, *p* = 0.45).

### 3.3. Biological Phenotype of COVID-19 Related Acute Pulmonary Embolism

COVID-19 patients at admission showed lower leukocytes count (including eosinophils, basophils, and lymphocytes counts: *p* < 0.005) and higher CRP values (*p* = 0.005) (Table 3). While platelets count (PC) and mean platelet volume (MPV) did not significantly differ between the two groups; the MPV/PC ratio was higher in coronavirus patients (*p* = 0.023). Likewise, abnormal coagulation parameters including activated partial thromboplastin time (aPTT) prolongation and higher fibrinogen were more frequently observed in COVID-19 patients (*p* < 0.005). D-Dimer level was lower at admission in COVID-19 patients (*p* = 0.047) whereas no gradual increase nor difference during hospital stay course were observed in the two subsets of patients. A dramatic increase in the incidence of lupus anticoagulant could be evidenced in COVID-19 patients in comparison with non-COVID-19 patients (82.4% vs. 12.5%, respectively, *p* < 0.001). During hospitalization, leukocytes peak, platelets peak, CRP, and fibrinogen peaks were significantly higher in patient with COVID-19 (*p* < 0.05). Renal injury was more frequent in COVID-19 patients with a higher creatinine peak. At discharge, no difference was found in the different blood tests performed.

### 3.4. Outcomes

Transfer to the ICU occurred more frequently in patients with COVID-19 (45.8% vs. 6.2%, *p* < 0.001) as did mechanical ventilation (40.7% vs. 6.2%, *p* < 0.001) (Table 1). There was a trend for a higher in-hospital death rate (15.3% vs. 4.9%, *p* = 0.07) and length of stay (15.5 ± 7.7 vs. 12.1 ± 13.4, *p* = 0.14) in COVID-19 patients in comparison with non-COVID-19 patients.

By univariate analysis, age, previous VTE, sPESI score, creatinine, platelets, leukocytes, CRP, fibrinogen (all values at peak during hospitalization), LA, and COVID-19 were significantly associated with the occurrence of transfer to the ICU in the study population (Table 4). Given the collinearity of LA and COVID-19, we built two models for multivariate analysis. In the first model, including all candidates’ predictors, except LA, creatinine at peak (hazard ratio (HR) 1.01, confidence interval (CI) 95% [1.00–1.02], *p* = 0.011), CRP at peak (HR 1.00, CI95% [1.00–1.01], *p* = 0.012) and COVID-19 (HR 4.19, CI95% [1.27–13.76], *p* = 0.018) were independent predictors of transfer to the ICU in patients with APE. In the second model, including all candidates’ predictors, except COVID-19, only CRP at peak (HR 1.01, CI95% [1.00–1.02], *p* = 0.029) was independently associated with transfer to the ICU.

## 4. Discussion

In this temporal analysis of 140 patients with acute pulmonary embolism, we observed a marked decline in non-COVID-19 APE during the first European wave of the pandemic. While there was similar thrombus load and location of the most proximal arterial branch involved in COVID-19 and non-COVID-19 patients, elevated markers of thrombosis and inflammation, and worse outcomes in patients with COVID-19 could be evidenced.

### 4.1. Insights from the Comparison between COVID-19 and Non-COVID-19 Patients

In line with previous studies reporting a decrease in the number of patients presenting to hospitals, because of emergency conditions, such as acute coronary syndrome [18,19,20], stroke [21] and APE [9], we observed a marked decline in non-COVID-19 APE between 1 March 2020 and 31 May 2020. Despite a significant increase in the use of CTPA imaging, we identified a 812% reduction in non-COVID-19 APE as compared with 2019. The reasons for this observed decline in traditional APE are likely multifactorial: stay-at-home message by governmental and healthcare institutions, fear of exposure to COVID-19 affected subjects at hospital admission.

Main features of COVID-19 patients with APE have been described in several studies [1,2,22]. As previously reported [22], we found a rare prevalence of the traditional risk factors in COVID-19 patients. History of venous thromboembolism and cancer, two strong predisposing factors of APE in hospitalized patients [23], were significantly less frequently encountered in patients with COVID-19 and none of them had a history of APE. Conversely, proportions of obesity and male sex were higher in COVID-19 patients. These factors have been established as predisposing factors of severe forms of COVID-19 [24]. Besides lung alveolar cells and vascular endothelium, ACE2 is also expressed in adipocytes and up-regulation of ACE2 has been described in obesity. Enhanced ACE2-mediated viral access and replication in adipose tissue was suggested as an important determinant of inflammatory burden [25].

Numerous studies have emphasized that COVID-19 is associated with coagulation parameters abnormalities [5,7]. In our experience, with respect to biological values measured in non-COVID-19 APE patients, the prothrombotic state observed in coronavirus patients was characterized by higher levels of fibrinogen and a marked increase of lupus anticoagulant incidence. Previous reports have emphasized a high incidence of positive lupus anticoagulant during COVID-19 [6]. While inflammation and CRP were known to interfere with some lupus anticoagulant tests, this high incidence persisted after adjustment for CRP levels [26]. It may be attributed to the strong inflammation and in severe cases to cytokine storms. The prolonged aPTT that we have observed in COVID-19 patients in comparison with non-COVID-19 patients could be explained in part by the higher incidence of lupus anticoagulant and thus should not be a limitation to the use of anticoagulation in the prevention and treatment of VTE, as previously proposed [6].

Surprisingly, we found lower D-Dimer level at admission among COVID-19 patients but no difference at peak with non-COVID-19 in contrast with the results published earlier by van Dam et al. [27]. Two hypothesizes can be raised to explain this difference. First, we could not exclude that a shift in the time course of the thromboembolism process occurred in patients with and without COVID-19 since the median delay between admission and APE diagnosis was longer in COVID-19 patients. In addition, it is likely that inflammatory cytokines such as tumor necrosis factor (TNF) can exert an inhibitory effect on the fibrinolytic system [28]. In line with this view, a previous study that included COVID-19 and non-COVID-19 patients hospitalized in ICU for acute respiratory distress syndrome (ARDS), found that D-Dimer was significantly lower in patients with COVID-19 [29].

Besides the prothrombotic state highlighted in COVID-19, previous studies have reported that thrombus described by CTPA was in the majority of cases segmental or subsegmental during COVID-19 related APE [27,30]. Some authors have suggested that localized immunothrombosis process could contribute to the development of a thrombus within the lung inflammation area [8]. In our study, the thrombus load was calculated from CTPA data using the Qanadli score. No difference could be evidenced between the two subsets of patients. Moreover, the distribution of thrombus localization was homogeneously distributed. As we extensively collected thrombus localization for all patients irrespective of the coronavirus status, we reported a systematic concordance of thrombus localization in subsegmental or segmental pulmonary artery and lung segments with COVID-19 related parenchymal injuries. Such findings support the importance of local pulmonary injuries and inflammation as key events in the thrombus growth [27].

In the field of immunothrombosis, circulating platelets have been recognized as the primary cells regulating hemostasis and thrombosis [31]. By expression of different receptors of immune response, such as toll-like receptors (TLR) or NOD-like receptors, platelets have the capacity to recognize viral pathogens [32]. The subsequent platelet activation leads to both inflammatory and prothrombotic response. A previous study showed that COVID-19 induced deep functional modifications in platelets [33]. In our study, while PC and MPV were no different, the MPV/PC ratio, a surrogate marker of platelet activation and function [34], was significantly increased in patients with COVID-19. Consistent with this analysis, an increased MPV/PC ratio has been reported as a risk factor of arterial and venous thrombosis [35].

### 4.2. Clinical Implications

On top of high reported incidence, APE is associated with higher transfer to the ICU and in-hospital death during COVID-19 [1,2]. In our study, COVID-19 patients showed worse prognosis, with more frequent ICU transfer and mechanical ventilation. APE prevention is of paramount importance leading to a prompt response of international medical societies to prevent thrombotic events in COVID-19 patients [36]. In line with previous reports [37,38], we found that a sizeable proportion of APE occurred in patients under preventive anticoagulation (35.6%) while it was very rare in patients without COVID-19 (6.2%). Some reports have found that therapeutic anticoagulation during hospital stay was associated with improved survival among COVID-19 patients [39,40]. The results from ongoing trials including the CORIMMUNO-COAG (NCT04344756), COVID-HEP (NCT04345848), and REMAP-CAP (NCT2735707) trials should further clarify this issue.

As recently highlighted in the American guidelines on VTE management in hospitalized patients, VTE risk assessment is a crucial issue [15]. Interestingly, while patients with and without COVID-19 did not share the same risk factors, the VTE prediction risk assessment via the Padua prediction score and the IMPROVE score did not differ between both patient types. It must be noted that all patients in our cohort had a Padua prediction score ≥ 4 translating a high risk of VTE. As no reliable, applicable, and well define scoring system currently exists to predict VTE in COVID-19, both the Padua and the IMPROVE score seem to well capture patients at risk of VTE, and might help to guide clinician decisions pending additional studies.

### 4.3. Study Limitations

We acknowledge several limitations. The retrospective nature of the study limits the generalizability of the findings. We cannot exclude selection bias in the comparison between patients with and without COVID-19. However, we used a control cohort of patients hospitalized for APE during the same period (1 March to 31 May) of the year 2019 to avoid any seasonality in APE causes. Both execution and frequency of DVT imaging tests were left to the discretion of the treating physician and this may have underestimated its prevalence. The low incidence of DVT in COVID-19 patients appears as a supplementary argument for an immunothrombosis process within the pulmonary vasculature in COVID-19 related APE rather than thrombus embolism, but his should be interpreted with caution with regard to different proportion of patients with DVT imaging. Finally, we did not use specific outcomes such as need for inotropic support. We used transfer to the ICU as primary outcome because it reflects both the need of hemodynamic support (inotropic support but also cardiocirculatory assistance) and respiratory support.

## 5. Conclusions

In addition to a marked decline of non-COVID-19 APE during the first wave of coronavirus pandemic in the year 2020, our temporal analysis of 140 patients with APE found a prothrombotic state, several markers of immunothrombosis, and worse outcomes in patients with COVID-19. These data reinforce the need to an efficient risk assessment and prevention of VTE during COVID-19.

## Figures and Tables

**Figure 1 jcm-10-02045-f001:**
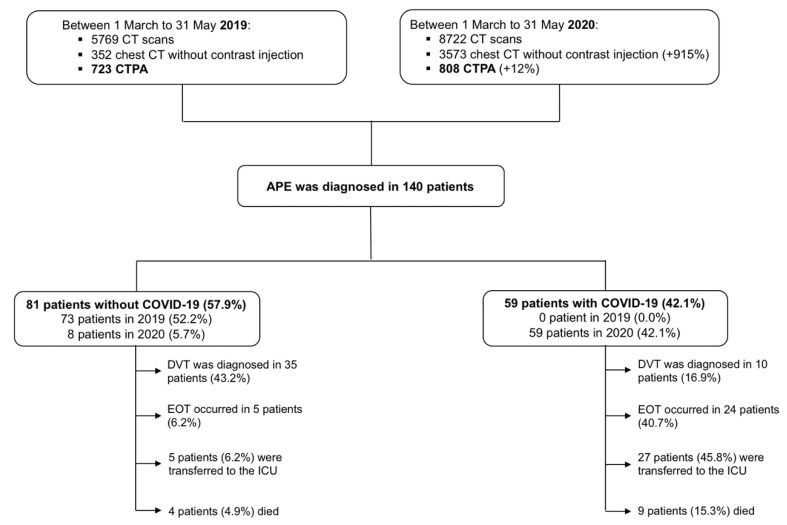
Flow-chart of the study.

**Figure 2 jcm-10-02045-f002:**
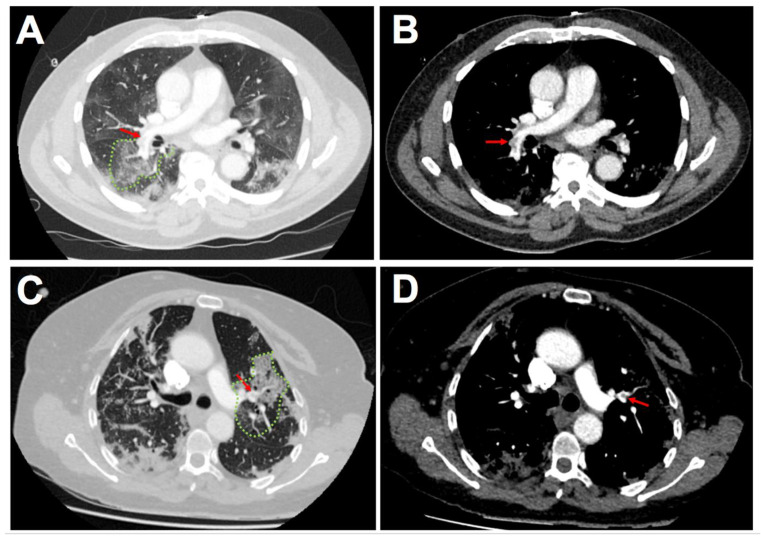
Examples of CTPA of COVID-19 patients with acute pulmonary embolism (lung windows: (**A**,**C**); mediastinal windows: (**B**,**D**)). Red arrows show lobar and segmental thrombus within the pulmonary vasculature. Green dots: areas with COVID-19 related parenchymal injuries. Abbreviations: COVID-19, coronavirus disease 2019; CTPA, computed tomography pulmonary angiography.

**Table 1 jcm-10-02045-t001:** Baseline characteristics.

	Patients with Pulmonary Embolism		*p* Value
	COVID-19 Negative (*n* = 81)	COVID-19 Positive (*n* = 59)	
**Demographic Characteristics**			
Age–y	70.2 ± 15.1	63.9 ± 14.4	0.014
Male–*n* (%)	41 (50.6)	41 (69.5)	0.037
**Cardiovascular risk factors**			
Obesity–*n* (%)	12 (14.8)	21 (35.6)	0.005
Hypertension–*n* (%)	51 (63.0)	30 (50.8)	0.169
Diabetes–*n* (%)	11 (13.6)	16 (27.1)	0.053
Dyslipidemia–*n* (%)	24 (29.6)	19 (32.8)	0.713
Smoking–*n* (%)	7 (8.6)	3 (5.1)	0.519
**Medical history**			
Previous VTE–*n* (%)	21 (25.9)	5 (8.5)	0.008
APE–*n* (%)	6 (7.4)	0 (0.0)	0.039
DVT–*n* (%)	17 (21.0)	5 (8.5)	0.059
Heart failure–*n* (%)	1 (1.2)	4 (6.8)	0.162
CKD *–*n* (%)	4 (4.9)	2 (3.4)	1.000
COPD–*n* (%)	5 (6.2)	2 (3.4)	0.699
Active cancer–*n* (%)	13 (16.0)	2 (3.4)	0.024
Cancer in remission–*n* (%)	4 (4.9)	7 (11.9)	0.202
**Medications before hospitalization**			
OAC–no. (%)	9 (11.1)	2 (3.4)	0.119
SAPT–no. (%)	18 (22.2)	7 (11.9)	0.125
DAPT–no. (%)	1 (1.2)	1 (1.7)	1.000
ACEi–no. (%)	7 (8.6)	10 (16.9)	0.190
ARBs–no. (%)	26 (32.1)	11 (18.6)	0.084
Beta-blocker–no. (%)	19 (23.5)	14 (23.7)	1.000
Statins–no. (%)	20 (24.7)	11 (18.6)	0.419
Oral contraceptives–no. (%)	2 (2.5)	0 (0.0)	0.509
**VTE risk assessment**			
Padua score ≥ 4–*n* (%) ^†^	81 (100)	59 (100)	1.000
IMPROVE score–*n* (%) ^‡^	1.9 ± 1.6	1.4 ± 0.9	0.060
**Thromboprophylaxis before VTE**			
None–*n* (%)	68 (84.0)	28 (47.5)	<0.001
Standard dose–*n* (%)	5 (6.2)	21 (35.6)	<0.001
Intermediate dose–*n* (%)	0 (0.0)	5 (8.5)	0.012
Therapeutic dose–*n* (%)	7 (8.6)	5 (8.5)	0.611
**Outcomes during hospitalization**			
Transfer to ICU–*n* (%)	5 (6.2)	27 (45.8)	<0.001
Need for mechanical ventilation–*n* (%)	5 (6.2)	24 (40.7)	<0.001
In-hospital death–*n* (%)	4 (4.9)	9 (15.3)	0.073
DVT–*n* (%) ^§^	35 (43.2)	10 (16.9)	0.001
Length of stay–days	12.1 ± 13.4	15.5 ± 7.7	0.142

Data are presented as mean ± standard deviation in case of any other indication. * Chronic kidney disease is defined by eGFR ≤ 60 mL/min/1.73 m^2^. ^†^ Venous thromboembolism risk was evaluated on admission to hospital via the Padua Prediction Score. ^‡^ Venous thromboembolism risk was also evaluated on admission to hospital via the Improve VTE risk score. ^§^ Imaging test for DVT was performed in 63 in patients without COVID-19 and in 14 patients with COVID-19. Abbreviations: ACEi, angiotensin converting enzyme inhibitor; ARBs, angiotensin-II receptor blockers; APE, acute pulmonary embolism; BMI, body mass index; CKD, chronic kidney disease; COPD, chronic obstructive pulmonary disease; COVID-19, coronavirus disease 2019; DAPT, dual antiplatelet therapy; DVT, deep vein thrombosis; ICU, intensive care medicine, OAC, oral anticoagulant; SAPT, single antiplatelet therapy; VTE, venous thromboembolism.

**Table 2 jcm-10-02045-t002:** Characteristics of acute pulmonary embolism stratified by the presence or absence of COVID-19.

	Patients with Pulmonary Embolism		*p* Value
	COVID-19 Negative (*n* = 81)	COVID-19 Positive (*n* = 59)	
**APE Severity**			
sPESI	0.83 ± 0.83	1.15 ± 0.76	0.019
Low risk–*n* (%)	27 (33.3)	10 (16.9)	0.034
Intermediate low risk–*n* (%)	36 (44.4)	27 (47.5)	0.735
Intermediate high risk–*n* (%)	17 (21.0)	19 (32.2)	0.171
High risk–*n* (%)	1 (1.2)	2 (3.4)	0.573
**APE localization**			
Sub-segmental–*n* (%)	9 (11.1)	6 (10.2)	0.859
Segmental–*n* (%)	20 (24.7)	21 (35.6)	0.162
Lobar–*n* (%)	26 (32.1)	15 (25.4)	0.391
Troncular–*n* (%)	26 (32.1)	17 (28.8)	0.677
Co-localization between segmental or subsegmental thrombus and COVID-19 related lung injuries–*n* (%)	-	27 (100)	-
**Thrombus load assessment**			
Qanadli score–IU	9.0 ± 7.4	8.1 ± 6.9	0.452

Data are presented as mean ± standard deviation in case of any other indication. Abbreviations: APE, acute pulmonary embolism; COVID-19, coronavirus disease 2019; sPESI, simplified pulmonary embolism severity index; IU, international units.

**Table 3 jcm-10-02045-t003:** Laboratory findings at admission, during hospitalization, and at discharge.

	Patients with Pulmonary Embolism		*p* Value
	COVID-19 Negative (*n* = 81)	COVID-19 Positive (*n* = 59)	
**At Admission**			
Leukocytes–×10^9^ per L	10.26 ± 3.48	9.00 ± 3.95	0.048
Neutrophils–×10^9^ per L	7.49 ± 3.26	7.14 ± 3.59	0.554
Eosinophils–×10^9^ per L	0.14 ± 0.14	0.03 ± 0.05	<0.001
Basophils–×10^9^ per L	0.05 ± 0.04	0.02 ± 0.02	<0.001
Lymphocytes–×10^9^ per L	1.59 ± 1.19	1.09 ± 0.53	0.003
Monocytes–×10^9^ per L	0.80 ± 0.34	0.69 ± 0.43	0.121
Hemoglobin–g/dL	12.4 ± 2.4	13.4 ± 2.2	0.008
Platelets–×10^9^ per L	266 ± 130	231 ± 99	0.088
MPV–fL	9.9 ± 10.5	10.3 ± 10.9	0.079
MPV/Platelets ratio–IU	4.1 ± 5.3	4.9 ± 7.4	0.023
Creatinine–µmol/L	77.1 ± 29.3	83.1 ± 45.0	0.349
eGFR–mL/min/1.73 m^2^	79 ± 23	84 ± 23	0.162
CRP–mg/L	63.7 ± 66.2	99.5 ± 78.1	0.005
Albumin–g/L	36.5 ± 8.2	34.7 ± 9.0	0.336
Troponin–ng/L	203.6 ± 543.9	384.0 ± 2124.3	0.506
BNP–pg/mL	218 ± 357	280 ± 616	0.497
PT–%	85 ± 15	85 ± 19	0.781
INR–IU	1.2 ± 0.4	1.3 ± 1.2	0.291
aPTT–IU	1.0 ± 0.2	1.2 ± 0.3	0.002
Fibrinogen–g/L	5.0 ± 1.6	6.2 ± 2.0	0.002
D-dimer–ng/mL	7389 ± 6736	4738 ± 5628	0.047
D-Dimer staging *–*n* (%)			0.023
<3 ULN	8 (15.7)	13 (31.7)	
3–6 ULN	10 (19.6)	11 (26.8)	
>6 ULN	33 (64.7)	17 (41.5)	
PO_2_–mmHg	85 ± 34	88 ± 35	0.684
PCO_2_–mmHg	36 ± 7	36 ± 9	0.912
PaO_2_/FiO_2_ ratio–IU	328 ± 122	266 ± 106	0.015
Lactate–mmol/L	1.4 ± 1.1	1.3 ± 0.5	0.730
**During Hospitalization**			
Leukocytes peak–×10^9^ per L	10.61 ± 4.15	12.77 ± 7.39	0.030
Hemoglobin nadir–g/dL	11.5 ± 2.3	12.4 ± 2.2	0.015
Platelets peak–×10^9^ per L	265 ± 129	343 ± 176	0.003
Creatinine peak–µmol/L	90.0 ± 39.8	120.7 ± 103.4	0.018
CRP peak–mg/L	74.1 ± 80.1	162.0 ± 106.5	<0.001
Fibrinogen peak–g/L	5.1 ± 1.9	7.3 ± 2.1	<0.001
D-dimer peak–ng/mL	7570 ± 6812	7168 ± 6595	0.771
D-Dimer staging *–*n* (%)			0.933
<3 ULN	9 (18.0)	7 (15.6)	
3–6 ULN	8 (16.0)	10 (22.2)	
>6 ULN	33 (66.0)	28 (62.2)	
Lupus anticoagulant positive–% *	1 (12.5)	28 (82.4)	<0.001
**At Discharge**			
Leukocytes–×10^9^ per L	15.62 ± 59.95	8.89 ± 5.38	0.450
Hemoglobin–g/dL	11.8 ± 2.5	13.4 ± 14.1	0.377
Platelets–×10^9^ per L	287 ± 138	336 ± 139	0.065
CRP–mg/L	51.7 ± 69.7	36.5 ± 49.7	0.222
Fibrinogen–g/L	5.3 ± 1.9	6.2 ± 2.0	0.218

Data are presented as mean ± standard deviation. * D-Dimer level at admission was available in 51 patients without COVID-19 and 41 patients with COVID-19. * Peak D-Dimer level was available in 50 patients without COVID-19 and 45 patients with COVID-19. * Lupus anticoagulant was tested in 8 patients without COVID-19 and in 34 patients with COVID-19. Abbreviations: aPTT, activated partial thromboplastin time ratio; BNP, B-type natriuretic peptide; CRP, C-reactive protein; eGFR, estimated glomerular filtration rate; FiO_2_, Fraction of inspired oxygen; INR, international normalized ratio; IU, international units; PCO_2_, partial pressure of carbon dioxide; PO_2_, partial pressure of oxygen; PT, Prothrombin time; VTE, venous thromboembolism.

**Table 4 jcm-10-02045-t004:** Univariate and multivariate analyses for occurrence of transfer to the intensive care unit in the study population.

Variables	Univariate Analysis		Multivariate Analysis			
	HR [95% CI]	*p*	First Model *		Second Model *	
			HR [95% CI]	*p*	HR [95% CI]	*p*
Age	0.96 [0.93–0.98]	0.006	0.96 [0.92–1.00]	0.057	1.01 [0.93–1.10]	0.660
BMI	1.05 [0.98–1.12]	0.130				
Previous VTE	0.114 [0.01–0.86]	0.035	0.11 [0.00–1.52]	0.101	0.10 [0.00–35.08]	0.108
Diabetes Mellitus	1.95 [0.77–0.49]	0.154				
CAD	0.91 [0.23–3.49]	0.893				
LV dysfunction	5.48 [0.87–34.37]	0.069				
Atrial Fibrillation	0.31 [0.03–2.56]	0.281				
COPD	0.54 [0.06–4.73]	0.585				
History of cancer	0.62 [0.19–1.99]	0.431				
Lack of thromboprophylaxis	0.48 [0.13–1.73]	0.265				
PE localization	0.89 [0.60–1.33]	0.599				
sPESI	1.91 [1.18–3.10]	0.008	1.73 [0.89–3.35]	0.102	2.54 [0.33–19.15]	0.365
Creatinine peak	1.01 [1.00–1.01]	0.001	1.01 [1.00–1.02]	0.011	1.02 [0.98–1.06]	0.260
Platelet peak	1.00 [1.00–1.00]	<0.001				
Leukocytes peak	1.12 [1.03–1.22]	0.004				
CRP peak	1.01 [1.00–1.01]	<0.001	1.00 [1.00–1.01]	0.012	1.01 [1.00–1.02]	0.029
Fibrinogen peak	1.99 [1.44–2.75]	<0.001				
D-Dimer peak	1.00 [1.00–1.00]	0.738				
Troponin peak	0.99 [0.98–1.00]	0.099				
LA	8.50 [1.60–45.12]	0.012			9.53 [0.38–238.92]	0.170
Qanadli score	0.97 [0.92–1.03]	0.395				
COVID-19	12.82 [4.53–36.27]	<0.001	4.19 [1.27–13.76]	0.018		

* First model included all candidates’ predictors except LA. Second model included all candidates’ predictors except COVID-19. Abbreviations: BMI, body mass index; CAD, coronary artery disease; CI, confidence interval; COPD, chronic obstructive pulmonary disease; COVID-19, coronavirus disease 2019; CRP, C-reactive protein; HR, hazard ratio; LA, lupus anticoagulant; LV, left ventricle; PE Pulmonary embolism; sPESI, simplified pulmonary embolism severity index; VTE, venous thromboembolism.

## Data Availability

The data presented in this study are available on request from the corresponding author.

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
