# Peer review of "Acute Pulmonary Embolism in Patients with and without COVID-19"

_jcm, 2021, doi:10.3390/jcm10102045_

Round 1
Reviewer 1 Report
Thank you for allowing me to review the study "Acute pulmonary embolism in patients with and without COVID-19" by Trimaille et al. This study attempts to investigate the prognosis and determinants of APE in COVID-19. Well written study with novel findings.
Please find below my suggestions:
Abstract: Line 16-17 needs to be re written Line 20-21 rephrase the dates sequentially Line 25 sPESI is an abbreviation with no explanation which is not allowed in the abstract Introduction: Lines 41-43: rewrite for grammar
Methods: Inclusion criteria: Only adults, if so, please include age criteria or range in methodology section Line 89: Expand abbreviations Why not have specific outcomes like need for inotropic support instead of ICU transfer Parametric for two groups is by t test and not ANOVA
Discussion: Line 242:Male sex
Author Response
Thank you for allowing me to review the study "Acute pulmonary embolism in patients with and without COVID-19" by Trimaille et al. This study attempts to investigate the prognosis and determinants of APE in COVID-19. Well written study with novel findings. Please find below my suggestions:
We would like to thank the Reviewer 1 for her/his positive appreciation of our work. Below are all the responses to these relevant queries.
Abstract: Line 16-17 needs to be re written
We have modified the sentence as follow:
Page 1, Lines 16-17: “Previous studies suggested an immunothrombosis instead of a thrombus embolism but the precise mechanisms remain unknown.”
Line 20-21 rephrase the dates sequentially
We have amended the sentence as follow:
Page 1, Lines 18-21: “We retrospectively included all consecutive patients with APE confirmed by computed tomography pulmonary angiography hospitalized at Strasbourg University Hospital from March 1 to May 31, 2019 and March 1 to May 31, 2020.”
Line 25 sPESI is an abbreviation with no explanation which is not allowed in the abstract
We have added the signification of this abbreviation:
Page 1, Lines 24-25: “COVID-19 patients showed a higher simplified pulmonary embolism severity index (sPESI) score”.
Introduction: Lines 41-43: rewrite for grammar
We have improved this part as follow:
Page 1, Lines 41-44: “Evidence from autopsy series reported endotheliitis [3], pulmonary vascular microthrombosis [4] and intense inflammation within the pulmonary vasculature. Laboratory features of hypercoagulability in COVID-19 include marked elevation of D-dimer and fibrinogen level, and the presence of lupus anticoagulant (LA) at high frequency [5–7].”
Methods: Inclusion criteria: Only adults, if so, please include age criteria or range in methodology section
We have added age criteria in the “Methodology” section of the main text:
Page2, Lines 58-61: “We retrospectively included all patients ≥18 years of age with CTPA confirmed APE admitted to general wards between March 1 and May 31 of the years 2019 and 2020.”
Line 89: Expand abbreviations
These abbreviations were explained:
Page 2, Lines 89-90: “Lesions severity of COVID-19 at CT was visually classified following the European Society of Radiology (ESR) / European Society of Thoracic Imaging (ESTI) guidelines.”
Why not have specific outcomes like need for inotropic support instead of ICU transfer
As depicted in the “Methods” section, we considered that ICU transfer reflects both the need of hemodynamic and respiratory support and better describes the prognosis of patients with acute pulmonary embolism. Moreover, the need for inotropic support is the main indication to ICU transfer. All patients with high-risk APE and who required inotropic support were transferred to the ICU in our center.
To clarify this point to the reader, we have added a sentence in the “Study limitations” section of the main text:
Page 11, Lines 348-350: “We used transfer to the ICU as primary outcome because it reflects both the need of hemodynamic support (inotropic support but also cardiocirculatory assistance) and respiratory support.”
Parametric for two groups is by t test and not ANOVA
This typo error was corrected:
Page 3, Lines 120-121: “Continuous variables were compared with the use of parametric (Student’s t test) or non-parametric Mann-Whitney tests as appropriate”.
Discussion: Line 239: Male sex
The sentence was amended as follow:
Page 11, Line 266: “Conversely, proportions of obesity and male sex were higher in COVID-19 patients.”
Reviewer 2 Report
Thank you for giving me the opportunity to review the manuscript-JCM-1198614, with the title "Acute pulmonary embolism in patients with and without COVID-19". This article addressed an important issue regarding the incidence, image pattern and predictor associated with COVID-19 related APE. I have some questions and suggestion as following:
1. The primary outcomes mentioned in abstract is different from mentioned in the part of introduction. Please clarify your primary outcome in the current study.
2. I think the greatest value in the current study is to compare the difference regarding biomarkers, image pattern and clinical score between COVID-19 and Non-COVID-19 associated APE. Therefore, reorganized the study aim and description in the part of introduction is necessary.
3. As shown in table 4, the percentage of patient required ICU admission and MV support was higher in the group of COVID-19 associated PE. This result may contribute from COVID-19 itself rather than PE. Therefore, Table 4 should be merged into table 1 as the demographic outcomes between COVID-19 positive & negative PE.
4. Since ICU admission is your primary outcome, therefore, predictors associated ICU admission in the cohort with APE is required to be analyzed. For example, Age, Obesity, Medical hx of previous VET, active cancer, thromboprophylaxis before VTE, peak of lab data (leukocytes, platelets, creatinine, CRP, fibrinogen, lupus anticoagulant positive…)
5. In the discussion part (Lin270-281), please make it more easily to read. Previous studies have reported that the distribution in COVID-19 related PE were frequently segmental or subsegmental distribution and the location of thrombus were mostly in the proximal part. This may contribute to local inflammation of lung according to previous research. However, there were no significant difference in the distribution and thrombus load according to table 2. How could you explaint it?
Author Response
Thank you for giving me the opportunity to review the manuscript-JCM-1198614, with the title "Acute pulmonary embolism in patients with and without COVID-19". This article addressed an important issue regarding the incidence, image pattern and predictor associated with COVID-19 related APE. I have some questions and suggestion as following:
We would like to thank the Reviewer 2 for her/his in-depth review of our work. Please find below our responses to these important questions.
- The primary outcomes mentioned in abstract is different from mentioned in the part of introduction. Please clarify your primary outcome in the current study.
The primary outcome of the study was transfer to the ICU, as described in the Abstract and “Methods” sections of the main text.
In the “Introduction” section, we described the primary objective of our study which was to compare the prevalence, clinical, biological and radiological determinants of COVID-19 and non COVID-19 APE admitted to general ward at our institution.
To clarify the objective of our study, we have amended the “Introduction” section as follow:
Page 2, Lines 49-52: “The objectives of our study were to compare the difference regarding clinical, biological and radiological characteristics of COVID-19 and non COVID-19 patients with APE admitted to general ward at our institution, and to assess the impact of COVID-19 in APE prognosis.”
- I think the greatest value in the current study is to compare the difference regarding biomarkers, image pattern and clinical score between COVID-19 and Non-COVID-19 associated APE. Therefore, reorganized the study aim and description in the part of introduction is necessary.
To better describe the objectives of the study, we have reorganized the “Introduction” section as follow:
Page 2, Lines 49-52: “The objectives of our study were to compare the difference regarding clinical, biological and radiological characteristics of COVID-19 and non COVID-19 patients with APE admitted to general ward at our institution, and to assess the impact of COVID-19 in APE prognosis.”
- As shown in table 4, the percentage of patient required ICU admission and MV support was higher in the group of COVID-19 associated PE. This result may contribute from COVID-19 itself rather than PE. Therefore, Table 4 should be merged into table 1 as the demographic outcomes between COVID-19 positive & negative PE.
We thank the Reviewer 2 for this pertinent suggestion. Accordingly, we have merged Table 4 into the Table 1.
- Since ICU admission is your primary outcome, therefore, predictors associated ICU admission in the cohort with APE is required to be analyzed. For example, Age, Obesity, Medical hx of previous VET, active cancer, thromboprophylaxis before VTE, peak of lab data (leukocytes, platelets, creatinine, CRP, fibrinogen, lupus anticoagulant positive…)
We thank the Reviewer 2 for this helpful suggestion. Accordingly, we have added univariate and multivariate analysis for predictors associated with transfer to the ICU in the study population.
Thus, we have added the description of this statistical analysis in “Methods” section of the main text:
Page 3, Lines 123-126: “Time to event was defined as the time from hospital admission to the date of transfer to the ICU, with patients censored at death or end of the study. The impact of COVID-19 on prognosis during APE was assessed using both univariate and multivariate Cox hazard model.”
We have also added the description of the results and the correspondent table:
Page 9, Lines 221-231: “By univariate analysis, age, previous VTE, sPESI score, creatinine, platelets, leukocytes, CRP, fibrinogen (all values at peak during hospitalization), LA and COVID-19 were significantly associated with the occurrence of transfer to the ICU in the study population (Table 4). Given the collinearity of LA and COVID-19, we have built two models for multivariate analysis. In the first model including all candidates’ predictors except LA, creatinine at peak (hazard ratio (HR) 1.01, confidence interval (CI) 95% [1.00-1.02], p=0.011), CRP at peak (HR 1.00, CI95% [1.00-1.01], p=0.012) and COVID-19 (HR 4.19, CI95% [1.27-13.76], p=0.018) were independent predictors of transfer to the ICU in patients with APE. In the second model including all candidates’ predictors except COVID-19, only CRP at peak (HR 1.01, CI95% [1.00-1.02], p=0.029) was independently associated with transfer to the ICU.”
Table 4. Univariate and multivariate analyses for occurrence of transfer to the intensive care unit in the study population.
Variables |
Univariate analysis |
Multivariate analysis |
||||
HR [95% CI] |
p |
First model* |
Second model* |
|||
HR [95% CI] |
p |
HR [95% CI] |
p |
|||
Age |
0.96 [0.93-0.98] |
0.006 |
0.96 [0.92-1.00] |
0.057 |
1.01 [0.93-1.10] |
0.660 |
BMI |
1.05 [0.98-1.12] |
0.130 |
|
|
|
|
Previous VTE |
0.114 [0.01-0.86] |
0.035 |
0.11 [0.00-1.52] |
0.101 |
0.10 [0.00-35.08] |
0.108 |
Diabetes Mellitus |
1.95 [0.77-0.49] |
0.154 |
|
|
|
|
CAD |
0.91 [0.23-3.49] |
0.893 |
|
|
|
|
LV dysfunction |
5.48 [0.87-34.37] |
0.069 |
|
|
|
|
Atrial Fibrillation |
0.31 [0.03-2.56] |
0.281 |
|
|
|
|
COPD |
0.54 [0.06-4.73] |
0.585 |
|
|
|
|
History of cancer |
0.62 [0.19-1.99] |
0.431 |
|
|
|
|
Lack of thromboprophylaxis |
0.48 [0.13-1.73] |
0.265 |
|
|
|
|
PE localization |
0.89 [0.60-1.33] |
0.599 |
|
|
|
|
sPESI |
1.91 [1.18-3.10] |
0.008 |
1.73 [0.89-3.35] |
0.102 |
2.54 [0.33-19.15] |
0.365 |
Creatinine peak |
1.01 [1.00-1.01] |
0.001 |
1.01 [1.00-1.02] |
0.011 |
1.02 [0.98-1.06] |
0.260 |
Platelet peak |
1.00 [1.00-1.00] |
<0.001 |
|
|
|
|
Leukocytes peak |
1.12 [1.03-1.22] |
0.004 |
|
|
|
|
CRP peak |
1.01[1.00-1.01] |
<0.001 |
1.00 [1.00-1.01] |
0.012 |
1.01 [1.00-1.02] |
0.029 |
Fibrinogen peak |
1.99 [1.44-2.75] |
<0.001 |
|
|
|
|
D-Dimer peak |
1.00 [1.00-1.00] |
0.738 |
|
|
|
|
Troponin peak |
0.99 [0.98-1.00] |
0.099 |
|
|
|
|
LA |
8.50 [1.60-45.12] |
0.012 |
|
|
9.53 [0.38-238.92] |
0.170 |
Quanadli score |
0.97 [0.92-1.03] |
0.395 |
|
|
|
|
COVID-19 |
12.82 [4.53-36.27] |
<0.001 |
4.19 [1.27-13.76] |
0.018 |
|
|
* First model included all candidates’ predictors except LA. Second model included all candidates’ predictors except COVID-19.
Abbreviations: BMI, body mass index; CAD, coronary artery disease; CI, confidence interval; COPD, chronic obstructive pulmonary disease; COVID-19, coronavirus disease 2019; CRP, C-reactive protein; HR, hazard ratio; LA, lupus anticoagulant; LV, left ventricle; PE Pulmonary embolism; sPESI, simplified pulmonary embolism severity index; VTE, venous thromboembolism
- In the discussion part (Lin270-281), please make it more easily to read. Previous studies have reported that the distribution in COVID-19 related PE were frequently segmental or subsegmental distribution and the location of thrombus were mostly in the proximal part. This may contribute to local inflammation of lung according to previous research. However, there were no significant difference in the distribution and thrombus load according to table 2. How could you explaint it?
In the “Discussion” section, we suggested that pulmonary interstitial inflammation secondary to lung cells invasion associated with endotheliitis following endothelial cells infection drive together an immuno-thrombosis process within the pulmonary vascular bed. The systematic concordance of thrombus localization in subsegmental or segmental pulmonary artery with the localization of COVID-19 related parenchymal injuries support this hypothesis. However, we did not find any difference in term of thrombus load as assessed by the Qanadli score or in the proportion of subsegmental and segmental thrombus between patients with and without COVID-19.
We amended this part as follow:
Page 11, Lines 294-305: “Besides the prothrombotic state highlighted in COVID-19, previous studies have reported that thrombus described by CTPA was in the majority of cases segmental or subsegmental during COVID-19 related APE [27,30]. Some authors have suggested that localized immunothrombosis process could contribute to the development of a thrombus within the lung inflammation area [8]. In our study, the thrombus load was calculated from CTPA data using the Qanadli score. No difference could be evidenced between the two subsets of patients. Moreover, the distribution of thrombus localization was homogeneously distributed. As we extensively collected thrombus localization for all patients irrespective of the coronavirus status, we reported a systematic concordance of thrombus localization in subsegmental or segmental pulmonary artery and lung segments with COVID-19 related parenchymal injuries. Such findings support the importance of local pulmonary injuries and inflammation as key events in the thrombus growth [27].”
Reviewer 3 Report
Remarks to the author:
In this study Trimaille et al has investigated the determinants and prognosis of acute pulmonary embolism (APE) during COVID-19. A retrospective study has been conducted in patients with APE (confirmed by computed tomography pulmonary angiography) who were hospitalized at Strasbourg University Hospital between March 1 and May 31 of the years 2019 and 2020. They have collected clinical, biological and imaging data from all the patients and evaluated. COVID-19 resulted in higher sPESI score and frequent transfer to ICU compared to non-COVID 19 patients included into the study. Moreover, in all COVID-19 patients, subsegmental or segmental APE, were localized in the areas with COVID-19 related lung injuries while showing a significant increase in inflammatory and prothrombotic biological markers. The authors have concluded that APE patients with COVID-19 have a prothrombotic state and a worse prognosis.
Specific comments:
- Materials and methods: The total the number of patients included into the study is not mentioned when describing the study population. Also, inclusion and exclusion criteria considered during patient recruitment to the study or either mentioning the fact that if it hasn’t been considered during data collection are applicable to provide clear idea about the settings and the study population.
- Please carefully check and expand the abbreviations at first mention (eg: VTE)
- Figure 2 shows the examples for CTPA of COVID-19 patients with acute pulmonary embolism. However, the panel labels for lung window and mediastinal window needed to appear as panel A and C, and panel B and D instead of A-C and B-D. Otherwise, A-C might be interpreted as A to C which include B as well.
- In conclusion, when referring to first wave of coronavirus pandemic, please refer to the year.
Author Response
In this study Trimaille et al has investigated the determinants and prognosis of acute pulmonary embolism (APE) during COVID-19. A retrospective study has been conducted in patients with APE (confirmed by computed tomography pulmonary angiography) who were hospitalized at Strasbourg University Hospital between March 1 and May 31 of the years 2019 and 2020. They have collected clinical, biological and imaging data from all the patients and evaluated. COVID-19 resulted in higher sPESI score and frequent transfer to ICU compared to non-COVID 19 patients included into the study. Moreover, in all COVID-19 patients, subsegmental or segmental APE, were localized in the areas with COVID-19 related lung injuries while showing a significant increase in inflammatory and prothrombotic biological markers. The authors have concluded that APE patients with COVID-19 have a prothrombotic state and a worse prognosis. Specific comments:
We thank the Reviewer 3 for her/his extensive review of our work. Below are our responses to these queries.
Materials and methods: The total the number of patients included into the study is not mentioned when describing the study population. Also, inclusion and exclusion criteria considered during patient recruitment to the study or either mentioning the fact that if it hasn’t been considered during data collection are applicable to provide clear idea about the settings and the study population.
According to the Reviewer’s concerns, we added details to the study population description and inclusion/exclusion criteria in the “Methods” section of the main text:
Page 2, Lines 58-61: “We retrospectively included all patients ≥18 years old with CTPA confirmed APE admitted to general wards between March 1 and May 31 of the years 2019 and 2020. Exclusion criteria were an unclear diagnosis of APE or the absence of APE on CTPA.”
In addition, the total number of patients screened and finally included in the study is now clearly stated in the “Results” section of the main text and showed in the Figure 1:
Page 3, Lines 131-133: “A total of 8722 chest CT were performed during the 2020 period (from March 1 to May 31), with an 915% and 12% increase respectively in chest CT (n=3573) and CTPA (n=808) compared with the equivalent period in 2019 (Figure 1).”
Please carefully check and expand the abbreviations at first mention (eg: VTE)
According to Reviewer’s suggestion, the abbreviations were screened and expanded throughout the manuscript.
Figure 2 shows the examples for CTPA of COVID-19 patients with acute pulmonary embolism. However, the panel labels for lung window and mediastinal window needed to appear as panel A and C, and panel B and D instead of A-C and B-D. Otherwise, A-C might be interpreted as A to C which include B as well.
We thank the Reviewer for her/his remark. Accordingly, the legend of the Figure 2 was amended:
Page 6, Lines 180-181: “Figure 2. Examples of CTPA of COVID-19 patients with acute pulmonary embolism (lung windows: A and C; mediastinal windows: B and D).”
In conclusion, when referring to first wave of coronavirus pandemic, please refer to the year.
We modified the conclusion to clarify the chronology:
Page 12-13, Lines 352-355: “In addition to a marked decline of non-COVID-19 APE during the first wave of coronavirus pandemic in the year 2020, our temporal analysis of 140 patients with APE found a prothrombotic state, several markers of immunothrombosis and worse outcomes in patients with COVID-19.”
Round 2
Reviewer 2 Report
My questions has been responded point by point from the author.
congratulations!!